# PNPLA1 is a transacylase essential for the generation of the skin barrier lipid ω-O-acylceramide

Yusuke Ohno[1], Nozomi Kamiyama[1], Shota Nakamichi[1] & Akio Kihara[1]

Lipids are the primary components of the skin permeability barrier, which is the body's most powerful defensive mechanism against pathogens. Acylceramide (ω-O-acylceramide) is a specialized lipid essential for skin barrier formation. Here, we identify *PNPLA1* as the long-sought gene involved in the final step of acylceramide synthesis, esterification of ω-hydroxyceramide with linoleic acid, by cell-based assays. We show that increasing triglyceride levels by overproduction of the diacylglycerol acyltransferase DGAT2 stimulates acylceramide production, suggesting that triglyceride may act as a linoleic acid donor. Indeed, the *in vitro* analyses confirm that PNPLA1 catalyses acylceramide synthesis using triglyceride as a substrate. Mutant forms of PNPLA1 found in patients with ichthyosis exhibit reduced or no enzyme activity in either cell-based or *in vitro* assays. Altogether, our results indicate that PNPLA1 is directly involved in acylceramide synthesis as a transacylase, and provide important insights into the molecular mechanisms of skin barrier formation and of ichthyosis pathogenesis.

[1] Laboratory of Biochemistry, Faculty of Pharmaceutical Sciences, Hokkaido University, Kita 12-jo, Nishi 6-chome, Kita-ku, Sapporo 060-0812, Japan. Correspondence and requests for materials should be addressed to A.K. (email: kihara@pharm.hokudai.ac.jp).

The body surface epidermis forms a permeability barrier, which has essential roles in protecting terrestrial animals from invasion of pathogens and harmful substances such as allergens and pollutants as well as from internal water loss. Accordingly, several cutaneous disorders—such as ichthyosis, atopic dermatitis, infectious diseases and dry skin—are characterized by alterations or defects of this skin barrier[1,2]. The principal compound family in the skin permeability barrier is lipids. Lipids form multi-layered structures (lipid lamellae) extracellularly in the stratum corneum, the outermost layer of epidermis, and their high hydrophobicity inhibits the invasion of external materials and water loss from inside the body[1,2].

To carry out this special, barrier-creation function, lipid lamellae contain unusual lipids. Approximately half of stratum corneum lipids are ceramide, which is the backbone of sphingolipids, and epidermis-specific ceramide species such as acylceramide ($\omega$-O-acylceramide) exist[3–6]. Acylceramide is especially important for skin barrier formation. Loss of acylceramide due to mutations in acylceramide synthesis genes leads to autosomal recessive congenital ichthyosis (ARCI) in humans and neonatal lethality in mouse models, where gene loss causes similar skin barrier defects as in humans[7–13]. Ichthyosis is characterized by dry, thickened and scaly skin, and the skin barrier defect in ARCI is the most severe among several types of ichthyoses[12].

The structure of acylceramide is quite unique. Although normal ceramides contain two hydrophobic chains, a long-chain base and a fatty acid (FA) with carbon chain-length of C16–24 (refs 6,14), acylceramide has an additional hydrophobic chain, linoleic acid (Fig. 1a). Furthermore, the chain-length of the FA moiety is extremely long (C28–C36) (refs 15,16). Therefore, acylceramide is one of the most hydrophobic lipids in mammalian bodies. The characteristic structure of acylceramide plays a pivotal role in organizing lipid lamellae[17]. In addition, acylceramide is also important as a precursor of protein-bound ceramide, which connects lipid lamellae and corneocytes[18].

Despite the physiological and pathological importance of acylceramide, the elucidation of the molecular mechanism by which it is created has not been completely resolved. Although recent studies have identified the genes involved in the acylceramide synthesis-specific reactions—such as the FA elongases *ELOVL1* and *ELOVL4*, which are involved in the elongation of very-long-chain (VLC) FAs (VLCFAs; ≥C21) to ultra-long-chain (ULC) FAs (ULCFAs; ≥C26), the FA $\omega$-hydroxylase *CYP4F22*, which hydroxylases the $\omega$-carbon of ULCFAs, and the ceramide synthase *CERS3*, which catalyses an amide bond formation between a long-chain base and a ULCFA (refs 7,9,11,13,19,20) (Fig. 1a)—the gene involved in the final step of acylceramide production, that is, ester bond formation between $\omega$-hydroxyceramide and linoleic acid, has not been identified. This means the molecular mechanism by which the skin barrier is created is still unclear.

Human cannot synthesize the acylceramide component linoleic acid. Therefore, linoleic acid is an essential FA that must be supplied from diet. Essential FA deficiency causes several skin symptoms including ichthyosis due to impairment of normal acylceramide production[3,18]. Large decreases in or loss of acylceramide due to mutations in the genes involved in acylceramide synthesis (such as *CERS3*, *CYP4F22* and *ELOVL4*) causes non-syndromic ARCI (*CERS3* and *CYP4F22*) as described above, or a syndromic form of ichthyosis (*ELOVL4*) (refs 5,8,10,12,21). Furthermore, decreases in acylceramide levels are also observed in atopic dermatitis patients[22,23].

Many ichthyosis-causative genes have been identified, and some of them have been shown to be involved in acylceramide synthesis (as described above) or protein-bound ceramide

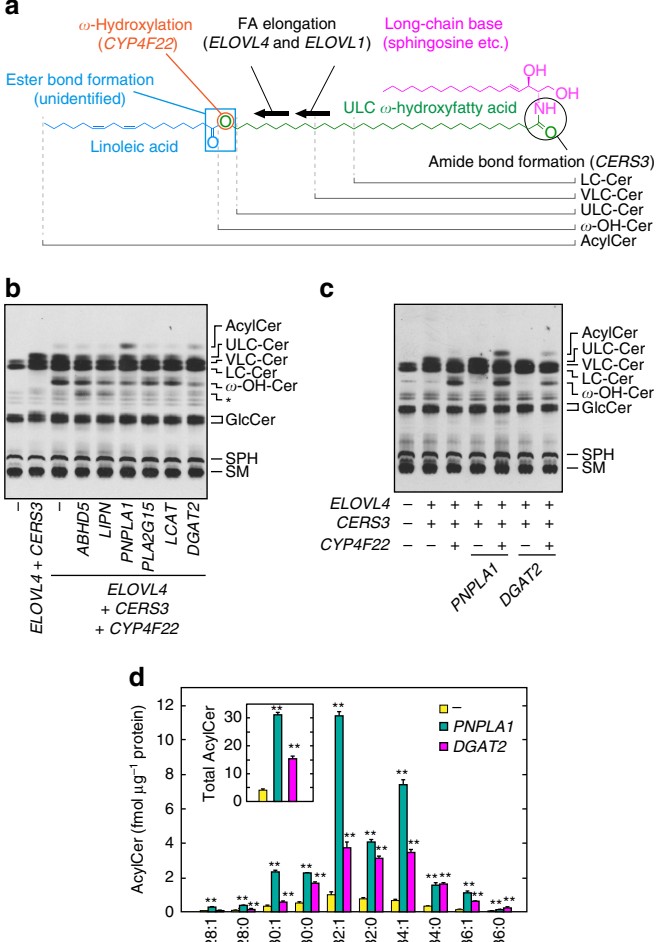

**Figure 1 | PNPLA1 is involved in acylceramide synthesis. (a)** Structure of acylceramide and genes involved in its synthesis. AcylCer, acylceramide; Cer, ceramide; $\omega$-OH-Cer, ULC $\omega$-hydroxyceramide. (**b,c**) HEK 293T cells were transfected with the plasmids encoding the indicated genes. Twenty-four hours after transfection, cells were labelled with [3H]sphingosine for 6 h at 37 °C in the presence of 10 μM linoleic acid. Lipids were extracted, separated by normal-phase TLC and detected by autoradiography. GlcCer, glucosylceramide; SM, sphingomyelin; SPH, sphingosine. The asterisk indicates an undetermined lipid, the production of which was not reproducible. Five independent experiments gave similar results, and a representative result is shown. (**d**) HEK 293T cells were transfected with an empty vector or the plasmid encoding *3xFLAG-PNPLA1* or *3xFLAG-DGAT2*, together with the plasmids encoding *3xFLAG-ELOVL4*, *3xFLAG-CERS3* and *3xFLAG-CYP4F22*. Three hours after transfection, 10 μM linoleic acid was added to the medium and incubated for 24 h at 37 °C. Lipids were extracted and subjected to LC-MS/MS analysis. Acylceramides were detected by MRM mode and quantified using MassLynx software. The graph depicts the amount of each acylceramide species with the indicated FA chain-length. The inset represents the total acylceramide levels. Values represent the means ± s.d.s of three independent experiments. Statistically significant differences compared to control cells are indicated (two-tailed Student's *t*-test; **$P < 0.01$).

production. However, there still remain several genes whose functions or pathogenic roles have not been revealed. For example, the functions of the ARCI-causative genes *NIPAL4* (NIPA-like domain-containing protein 4) and *PNPLA1* (patatin-like phospholipase domain-containing protein 1) are currently unclear[12,24,25].

In the present study, we aim to identify the missing gene responsible for the final step of acylceramide production (ester

bond formation between $\omega$-hydroxyceramide and linoleic acid). Our results indicate that *PNPLA1* encodes the transacylase that catalyses acylceramide production using a triglyceride (TG) as the donor of the substrate linoleic acid. Thus, our findings provide important insights into the molecular mechanism of acylceramide production and into the function and pathogenic role of the ichthyosis-causative gene *PNPLA1*.

## Results

**PNPLA1 is involved in acylceramide production.** To identify the acyltransferase or transacylase involved in acylceramide production, a proper assay system that can detect its activity or product is necessary. However, the prior lack of such assay systems meant there was no way of identifying the responsible acyltransferase/transacylase for a long time. Cell-based assays had been unsuccessful, since most mammalian cells cannot produce ULC $\omega$-hydroxyceramides, the precursors of acylceramides. However, we recently established a cell system that produces ULC $\omega$-hydroxyceramides by overproducing the FA elongase ELOVL4, the ceramide synthase CERS3 and the FA $\omega$-hydroxylase CYP4F22 in HEK 293T cells[13], opening the door to identify the acyltransferase/transacylase of interest.

In this cell system, when lipids prepared from HEK 293T cells labelled with [3H]sphingosine, the sphingolipid precursor, were separated by normal phase thin layer chromatography (TLC), only long-chain (LC; C11–C20) and VLC ceramides were detected as ceramide species (Fig. 1b). Overexpression of CERS3 and ELOVL4 caused cells to produce ULC ceramides, and further co-overproduction of CYP4F22 leads to production of ULC $\omega$-hydroxyceramides, as we have described previously[13] (Fig. 1b). Using this cell system, we examined the involvement of several candidate genes in acylceramide production. The selected candidate genes were *ABHD5* ($\alpha/\beta$ hydrolase domain containing 5)/CGI-58 (comparative gene identification-58), *LIPN* (lipase, family member N), *PNPLA1*, *PLA2G15* (phospholipase A₂, group XV)/LLPL (lecithin:cholesterol acyltransferase-like lysophospholipase), *LCAT* (lecithin:cholesterol acyltransferase) and *DGAT2* (diacylglycerol O-acyltransferase 2). *ABHD5* is a causative gene of Chanarin–Dorfman syndrome (also known as neutral lipid storage disease with ichthyosis (NLSD-I)), an autosomal recessive disease accompanied by ichthyosis, steatosis and other symptoms[26]. *LIPN* and *PNPLA1* are ARCI-causative genes[12,24,25,27], but their roles in skin barrier formation have not yet been revealed. Reasoning that since ABHD5, LIPN and PNPLA1 contain phospholipase/hydrolase domains and some proteins containing such domains act as acyltransferases or transacylases[28], we chose them as candidates for acylceramide synthetic acyltransferases/transacylases. *PLA2G15* has been implicated in the synthesis of 1-*O*-acylceramide, another type of acylceramide with unknown function[29]. We examined the possibility that PLA2G15 (and its homologue LCAT) is also involved in acylceramide ($\omega$-*O*-acylceramide) synthesis. *DGAT2* encodes a diacylglycerol acyltransferase, and *Dgat2* knockout mice exhibit a skin-barrier-defect phenotype[30].

Expression of PNPLA1 caused acylceramide production, while that of ABHD5, LIPN, PLA2G15 or LCAT had no effect (Fig. 1b). Expression of DGAT2 also caused acylceramide synthesis, albeit weakly. DGAT2 is involved in TG synthesis[30,31], suggesting that TGs are somehow involved in acylceramide production, perhaps as a linoleic acid donor. To discriminate whether the produced acylceramide was $\omega$-*O*-acylceramide or 1-*O*-acylceramide, we next performed the [3H]sphingosine labelling assay in the presence or absence of the FA $\omega$-hydroxylase CYP4F22. Acylceramide production induced by PNPLA1 and by DGAT2 expression was CYP4F22-dependent in both cases (Fig. 1c): in

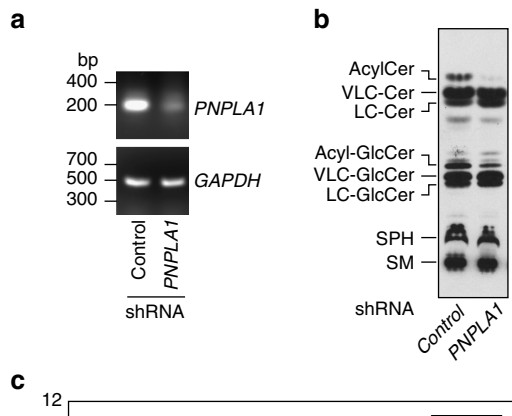

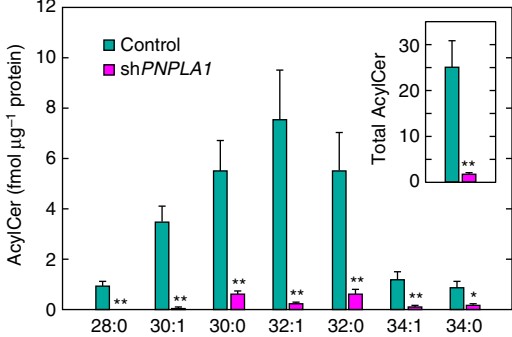

**Figure 2 | Knockdown of *PNPLA1* causes impairment of acylceramide production.** Keratinocytes were infected with lentivirus harbouring control shRNA or sh*PNPLA1* and were differentiated for 7 days. (**a**) Total RNAs prepared from differentiated keratinocytes were subjected to RT-PCR using primers specific for *PNPLA1* and *GAPDH*. Uncropped scanned photographs of the electrocataphoresis gels are provided as Supplementary Fig. 2. (**b**) Cells were labelled with [3H]sphingosine for 6 h at 37 °C in the presence of 10 μM linoleic acid. Lipids were extracted, separated by normal-phase TLC and detected by autoradiography. AcylCer, acylceramide; Cer, ceramide; Acyl-GlcCer, acyl-glucosylceramide; GlcCer, glucosylceramide; SPH, sphingosine; SM, sphingomyelin. Two independent experiments gave similar results, and a representative result is shown. (**c**) Lipids were extracted and subjected to LC-MS/MS analysis. Acylceramides were detected in the MRM mode and quantified using MassLynx software. The graph depicts the amount of each acylceramide species with the indicated FA chain-length. The inset represents the total acylceramide levels. Values represent the means ± s.d.s of three independent experiments. Statistically significant differences compared to control cells are indicated (two-tailed Student's *t*-test; *$P < 0.05$; **$P < 0.01$).

other words, it was FA $\omega$-hydroxylation-dependent, indicating that the produced acylceramide was $\omega$-*O*-acylceramide. Liquid chromatography–tandem mass spectrometry (LC-MS/MS) analysis revealed that acylceramides with carbon chain-length of C28–36, mainly C30–36, were produced by expression of PNPLA1 and DGAT2 (Fig. 1d). PNPLA1 and DGAT2 expression increased acylceramide levels by 7.6- or 3.8-fold compared to control, respectively (Fig. 1d, inset).

To confirm that PNPLA1 is indeed involved in acylceramide production, we performed knockdown analysis using human keratinocytes and a lentiviral vector-encoding shRNA system. Infection with a lentivirus bearing the shRNA specific for *PNPLA1* (sh*PNPLA1*) caused a large decrease in *PNPLA1* mRNA levels compared with the control (Fig. 2a). [3H]Sphingosine labelling experiments revealed that differentiated keratinocytes treated with control shRNA generated acylceramide and its derivative acyl-glucosylceramide (Fig. 2b). Treatment of keratinocytes with sh*PNPLA1* caused large decreases in

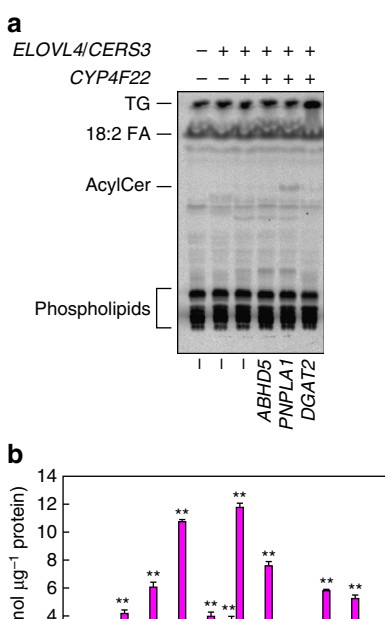

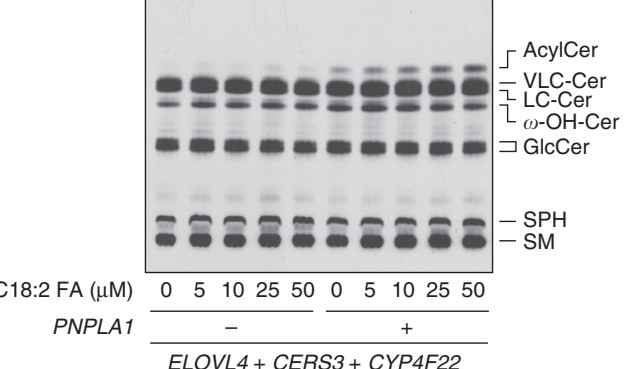

**Figure 4 | Linoleic acid concentration-dependent acylceramide production by PNPLA1.** HEK 293T cells were transfected with an empty vector or the plasmid encoding *3xFLAG-PNPLA*, together with the plasmids encoding *3xFLAG-ELOVL4*, *3xFLAG-CERS3* and *3xFLAG-CYP4F22*. Twenty-four hours after transfection, cells were treated with the indicated concentration of linoleic acid and labelled with [3H]sphingosine for 6 h at 37 °C. Lipids were extracted, separated by normal-phase TLC and detected by autoradiography. 18:2 FA, linoleic acid; AcylCer, acylceramide; Cer, ceramide; GlcCer, glucosylceramide; SM, sphingomyelin; SPH, sphingosine; $\omega$-OH-Cer, ULC $\omega$-hydroxyceramide. Four independent experiments gave similar results, and a representative result is shown.

**Figure 3 | Overexpression of DGAT2 but not PNPLA1 causes an increase in TG levels.** (**a**) HEK 293T cells transfected with plasmids encoding the indicated genes were labelled with [14C]linoleic acid for 4 h at 37 °C. Lipids were extracted, separated by normal-phase TLC and detected by autoradiography. 18:2 FA, linoleic acid; AcylCer, acylceramide. Four independent experiments gave similar results, and a representative result is shown. (**b**) HEK 293T cells were transfected with an empty vector or the plasmid encoding *3xFLAG-PNPLA1* or *3xFLAG-DGAT2*. Twenty-four hours after transfection, lipids were extracted and subjected to LC-MS/MS analysis. TGs containing linoleic acid were detected by MRM mode and quantified using MassLynx software. The graph depicts the amount of each TG species containing linoleic acid (18:2) and FAs with the indicated chain-length and degree of unsaturation. For example, 30:1 means that the sum of chain-length and degree of unsaturation of the two FA chains other than linoleic acid are 30 and 1, respectively. Values represent the means ± s.d.s of six (for control) or three (for *PNPLA1* and *DGAT2*) independent experiments. Statistically significant differences compared to control cells are indicated (two-tailed Student's *t*-test; **$P < 0.01$).

acylceramide/acyl-glucosylceramide levels. LC-MS/MS analysis confirmed that acylceramide levels were decreased by sh*PNPLA1* treatment, irrespective of chain-lengths (Fig. 2c). Total acylceramide levels in sh*PNPLA1*-treated cells were 7.1% of those in control shRNA-treated cells (Fig. 2c, inset). These results indicate that PNPLA1 is indeed involved in acylceramide production.

**PNPLA1 is not involved in TG metabolism**. Stimulation of acylceramide synthesis by DGAT2 suggests that TGs are involved in acylceramide production. We next examined the possibility that PNPLA1 also stimulates acylceramide synthesis through an increase in TG levels by [14C]linoleic acid labelling assay. DGAT2 expression increased TG levels as expected, whereas PNPLA1 had no effect (Fig. 3a). We next measured linoleic acid-containing TG levels by LC-MS/MS. The amounts of all of the TGs examined were increased by DGAT2 overexpression (Fig. 3b). In contrast, PNPLA1 again had almost no effect, whereas slight increases were observed for some TG species with shorter chain-lengths for

unknown reasons. However, these slight changes might not be able to cause the acylceramide increase. Thus, these results suggest that PNPLA1 does not affect acylceramide production through increasing TG levels.

PNPLA1 belongs to the patatin-like phospholipase domain-containing protein (PNPLA) family. The PNPLA family members are known to exhibit phospholipase, TG hydrolase or transacylase activity, or a combination[28]. Considering these functions of the PNPLA family members, PNPLA1 was expected to be involved in acylceramide production as a TG hydrolase/phospholipase supplying linoleic acid from TG/phospholipids to an unknown acyltransferase, or as a transacylase catalysing acylceramide production directly using the linoleic acid in TGs as a substrate. If the former possibility were true, an increase in cellular linoleic acid levels caused by adding linoleic acid exogenously might bypass the otherwise required step of PNPLA1 in acylceramide production. However, addition of linoleic acid in the medium did not cause an increase in acylceramide in the absence of PNPLA1 (Fig. 4). On the other hand, linoleic acid stimulated acylceramide synthesis in a dose-dependent manner in cells overproducing PNPLA1 (Fig. 4). These results suggest that the role of PNPLA1 in acylceramide synthesis is not to supply linoleic acid as a TG hydrolase/phospholipase, and instead that PNPLA1 may be directly involved in acylceramide synthesis. The conclusion—that it is unlikely PNPLA1 acts as a TG hydrolase—is consistent with the MS data, which showed that PNPLA1 expression did not reduce TG levels (Fig. 3b).

**PNPLA1 is a transacylase using TG as a substrate**. To prove that PNPLA1 directly catalyses acylceramide production as a transacylase using TG as a substrate, we performed *in vitro* assays. For this purpose, PNPLA1 was translated using a wheat germ cell-free translation system. Since PNPLA1 is a membrane protein[32], we added liposomes to the translation reaction mixture. Recently, several membrane proteins have successfully been inserted directly into the lipid bilayer of liposomes by similar cell-free translation systems[33–35]. After translation of *PNPLA1* mRNA, the resulting proteoliposomes were recovered by centrifugation and

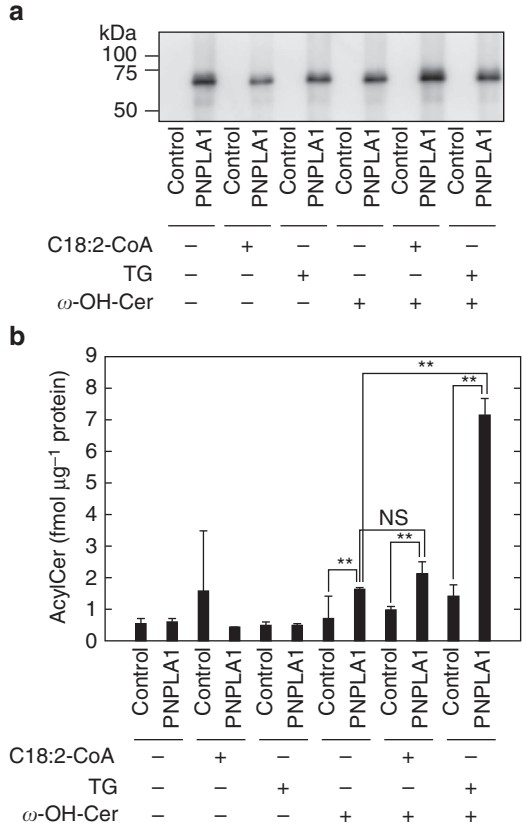

**Figure 5 | PNPLA1 catalyses acylceramide synthesis via transacylation.**
SP6 RNA polymerase was used *in vitro* to transcribe genes from control plasmid (pEU-E01-T1R1) or plasmid encoding wild-type *PNPLA1* (pEU-E01-3xFLAG-PNPLA1). The resulting mRNAs were then incubated with wheat germ lysates and phosphatidylcholine-based liposomes containing linoleoyl-CoA (C18:2-CoA), TG, and C30:0 ω-hydroxyceramide (ω-OH-Cer) as indicated, producing proteoliposomes. Proteoliposomes were then recovered by centrifugation. (**a**) Proteins associated with the prepared proteoliposomes were subjected to immunoblotting with anti-FLAG-antibody. An uncropped scan of the blot is provided as Supplementary Fig. 3. (**b**) An *in vitro* acylceramide synthesis assay was performed by incubating proteoliposomes for 1 h at 37 °C. Lipids were extracted, and the C30:0 acylceramide EOS was quantified by LC-MS/MS analysis. Values represent the means ± s.d.s of three independent experiments. Statistically significant differences are indicated (two-tailed Student's *t*-test; **$P < 0.01$). NS, not significant.

used for further analyses. We confirmed production of PNPLA1 by immunoblotting (Fig. 5a). Then, we subjected the proteoliposomes to an *in vitro* acylceramide synthesis assay, where linoleoyl-CoA (C18:2–CoA), TG and ω-hydroxyceramide were used as substrates in different combinations. The highest activity was observed when proteoliposomes containing PNPLA1, TG and ω-hydroxyceramide were used (Fig. 5b). Diglyceride levels were also increased (Supplementary Fig. 1), indicating that the linoleic acid portion of TG was transferred to ω-hydroxyceramide to produce acylceramide. Low levels of acylceramides were also produced by PNPLA1 in the presence of ω-hydroxyceramide alone (without exogenous TG). This is probably due to a supply of TG from the wheat germ lysates used in the cell-free translation system, as we confirmed by LC-MS/MS analysis. Inclusion of linoleoyl-CoA (C18:2–CoA) in the proteoliposomes containing PNPLA1 and ω-hydroxyceramide did not cause a further increase in acylceramide levels. These results indicate that PNPLA1 is a

transacylase using TG as a substrate rather than an acyltransferase using linoleoyl-CoA.

**Correlation between PNPLA1 activity and ichthyosis pathology.**
*PNPLA1* is one of the genes known to cause ARCI (refs 24,25). Two missense mutations, which cause amino acid substitution (A34T or A59V), and one nonsense mutation (E131X) have been found in the *PNPLA1* of ichthyosis patients[24,25]. The mutated residues (Ala34 and Ala59) are located in the patatin domain. We expressed wild type and mutant forms of PNPLA1 in HEK 293T cells together with ELOVL4, CERS3 and CYP4F22 and examined the expression and acylceramide production activities of these mutant PNPLA1 proteins. The point mutants PNPLA1 A34T and A59V were expressed at equivalent levels to the wild-type protein (Fig. 6a). [³H]Sphingosine labelling assay revealed that their activities were decreased to ~20% of wild-type protein activity (Fig. 6b). The nonsense mutant protein PNPLA1 E131X was detected as a truncated protein of 14 kDa (Fig. 6a). The acylceramide levels in PNPLA1 E131X-producing cells were indistinguishable from those in vector-transfected cells (Fig. 6b), indicating that the truncated PNPLA1 protein (E131X) had no acylceramide production activity.

Next, we directly measured the transacylase activities of the ichthyosis mutants of PNPLA1 *in vitro*. PNPLA1 mutants were properly expressed by the cell-free translation system in a manner equivalent to the wild-type protein (Fig. 6c). An acylceramide synthesis assay in the presence of TG and ω-hydroxyceramide revealed that both of the point mutants (PNPLA1 A34T and A59V) exhibited reduced activities compared to wild type. The truncated mutant E131X had no activity (Fig. 6d), consistent with the results obtained from the cell-based assay (Fig. 6b). These results show a clear relationship between PNPLA1 activity and ichthyosis pathology. Although it has been unclear why *PNPLA1* mutations cause ichthyosis, our results suggest that decreases in acylceramide levels are the cause of skin barrier defects and lead to ichthyosis.

**Discussion**
Over 30 years have passed since acylceramide and acyl-glucosylceramide were discovered and their structures determined[36,37]. However, the synthetic genes of acylceramide remained unclear for a long time. There had long been only limited knowledge about candidate genes involved in acylceramide production. However, recent determination of ichthyosis-causative genes gave a clue to the identification of acylceramide synthetic genes. For example, the ichthyosis-causative genes *CERS3*, *ELOVL4* and *CYP4F22* are all involved in acylceramide production[8–10,13,20–21]. In addition, we recently succeeded in establishing a cell system to produce ULC ω-hydroxyceramide, the substrate of acylceramide, by overproducing CERS3, ELOVL4 and CYP4F22 in HEK 293T cells[13]. Using this system again here, we revealed that the ARCI-causative gene *PNPLA1* is involved in acylceramide production (Fig. 1b–d). Furthermore, we demonstrated that PNPLA1 catalyses the transacylation of the linoleic acid portion of TG to ULC ω-hydroxyceramide for acylceramide production *in vitro* (Fig. 5). Murakami and his colleagues reached the same conclusion using *Pnpla1* knockout mice[38]. Their mice exhibited neonatal lethality due to skin barrier defects. Acylceramide was not produced in *Pnpla1* knockout mice, but its substrate ULC ω-hydroxyceramide accumulated. During the preparation of our manuscript, similar *in vivo* results were also reported by another group[39]. In that study, decreases in acylceramide levels in the differentiated keratinocytes from an ARCI patient were also reported. Taking our cell-assay-based and biochemical *in vitro*

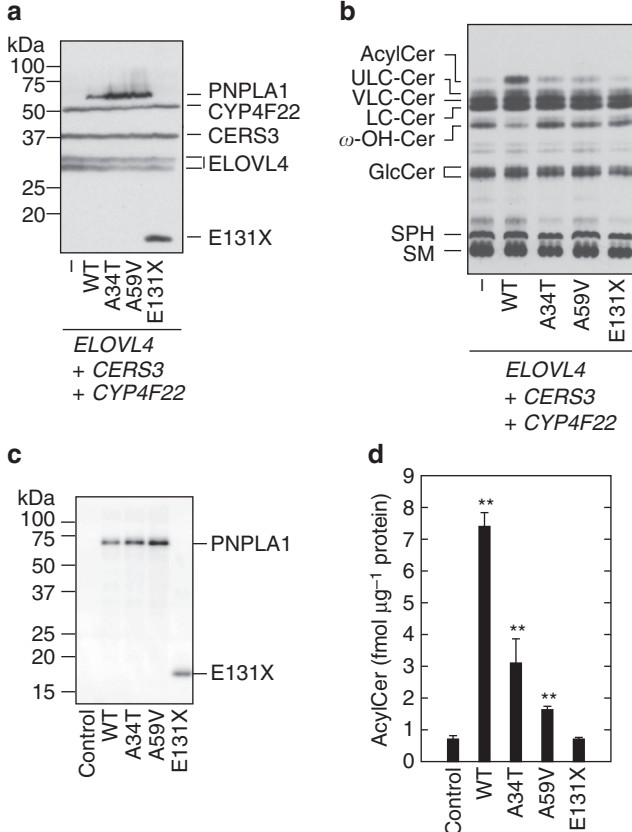

**Figure 6 | PNPLA1 activity is impaired in ichthyosis-causative mutants.**
(**a,b**) HEK 293T cells were transfected with the plasmids encoding *3xFLAG-ELOVL4*, *3xFLAG-CERS3*, *3xFLAG-CYP4F22* and *3xFLAG-PNPLA1* (wild type (WT) or mutant) as indicated. (**a**) Total cell lysates were separated by SDS-PAGE and subjected to immunoblotting with anti-FLAG antibody. (**b**) Cells were labelled with [$^3$H]sphingosine in the presence of 25 μM linoleic acid for 6 h at 37 °C. Lipids were extracted, separated by normal-phase TLC and detected by autoradiography. AcylCer, acylceramide; Cer, ceramide; GlcCer, glucosylceramide; SM, sphingomyelin; SPH, sphingosine; ω-OH-Cer, ULC ω-hydroxyceramide. Three independent experiments gave similar results, and a representative result is shown. (**c,d**) Genes on control plasmid (pEU-E01-T1R1) or plasmid encoding wild type (WT; pEU-E01-3xFLAG-PNPLA1) or mutant *PNPLA1* (for A34T, pEU-E01-3xFLAG-PNPLA1(A34T); for A59V, pEU-E01-3xFLAG-PNPLA1(A59V); or for E131X, pEU-E01-3xFLAG-PNPLA1(E131X)) were transcribed *in vitro* using SP6 RNA polymerase. The resulting mRNAs were then incubated with wheat germ lysates and phosphatidylcholine-based liposomes containing TG and C30:0 ω-hydroxyceramide. (**c**) Proteins associated with the resulting proteoliposomes were separated by SDS-PAGE, followed by immunoblotting with anti-FLAG-antibody. (**d**) An *in vitro* acylceramide synthesis assay was performed by incubating proteoliposomes for 1 h at 37 °C. Lipids were extracted, and the C30:0 acylceramide EOS was quantified by LC-MS analysis. Values represent the means ± s.d.s of three independent experiments. Statistically significant differences compared to control are indicated (two-tailed Student's *t* test; **$P < 0.01$).

results with the *in vivo* results from the two groups, we conclude that PNPLA1 plays an essential role in the final step of acylceramide production, esterification of ULC ω-hydroxyceramide with linoleic acid.

In addition to PNPLA1, overproduction of the diacylglycerol acyltransferase DGAT2 also caused an increase in acylceramide levels in our cell system, although the effect was weaker than PNPLA1 (Fig. 1b–d). Since DGAT2 is involved in TG synthesis[30], it is likely that an increase in the levels of TG, as the substrate of

PNPLA1, by overproduction of DGAT2 (Fig. 3) indirectly enhances acylceramide production. *Dgat2* knockout mice exhibit a skin-barrier-defect phenotype[30], although the mechanism remains unclear. From our results, we speculate that the decreased TG causes impairment of acylceramide synthesis, leading to the skin barrier defect.

*PNPLA1* is known as an ARCI-causative gene[24,25]. However, it was previously unclear by what mechanism *PNPLA1* mutations cause ARCI. In the present study, we revealed that the PNPLA1 mutant proteins exhibited weak or no acylceramide production activity (Fig. 6). Thus, our results suggest that the pathology of ARCI associated with *PNPLA1* mutation is caused by reduced acylceramide levels. Similar correlations among ichthyosis pathology, enzyme activities and acylceramide levels have been observed for ARCI caused by *CERS3* and *CYP4F22* mutations as well[10,13].

Our results indicate that PNPLA1 is the transacylase that acts at the final step of acylceramide production, esterification between ULC ω-hydroxyceramide and linoleic acid. Since acylceramide is essential to maintain skin barrier integrity, our findings constitute important information by which we can understand the molecular mechanisms behind skin barrier formation. At present, there are no therapeutic agents for the causal treatment of ichthyosis or atopic dermatitis. Elucidation of the molecular mechanisms behind skin barrier formation may lead to the development of such new therapeutic medicines.

## Methods

**Cell culture and transfection.** HEK 293T cells and Lenti-X 293T cells (Takara Bio, Shiga, Japan; catalogue number, 632180) were cultured in Dulbecco's modified Eagle's medium (DMEM; Sigma, St Louis, MO, USA) supplemented with 10% fetal bovine serum (FBS), 100 U ml$^{-1}$ penicillin and 100 μg ml$^{-1}$ streptomycin, and were grown in dishes pre-coated with 0.1 mg ml$^{-1}$ collagen (Cellmatrix type I-P; Nitta Gelatin, Osaka, Japan). Transfections were performed using ViaFect Transfection Reagent (Promega, Madison, WI, USA), according to the manufacturer's instructions. Human keratinocytes (CELLnTEC, Bern, Switzerland; catalogue number, HPEKp) were cultured in CnT prime Epidermal Keratinocyte Medium (CELLnTEC). Differentiation was induced by incubating the cells with CnT-Prime 3D Barrier Medium (CELLnTEC).

**Plasmids.** The mammalian expression vector pCE-puro 3xFLAG-1 is derived from the pCE-puro vector[40] and designed for N-terminal 3xFLAG-tagged protein production[41]. Human *ABHD5*, *LIPN*, *PNPLA1*, *PLA2G15*, *LCAT* and *DGAT2* genes were amplified by reverse transcription (RT)-PCR using their respective forward (-F) and reverse (-R) primers listed in Supplementary Table 1. The PCR fragments were first cloned into pGEM-T Easy Vector (Promega) and then transferred to the pCE-puro 3xFLAG-1 vector, producing the pCE-puro 3xFLAG-ABHD5, pCE-puro 3xFLAG-LIPN, pCE-puro 3xFLAG-PNPLA1, pCE-puro 3xFLAG-PLA2G15, pCE-puro 3xFLAG-LCAT and pCE-puro 3xFLAG-DGAT2 plasmids. The pCE-puro 3xFLAG-ELOVL4, pCE-puro 3xFLAG-CERS3 and pCE-puro 3xFLAG-CYP4F22 plasmids encoding 5′-terminally *3xFLAG*-tagged *ELOVL4*, *CERS3* and *CYP4F22*, respectively, have been described previously[11,13]. Ichthyosis mutations were introduced into the *PNPLA1* gene using the QuikChange Site-Directed Mutagenesis Kit (Agilent Technologies, Santa Clara, CA, USA), using the primers described in Supplementary Table 1. For *in vitro* transcription, wild-type and mutant versions of *PNPLA1* tagged with *3xFLAG* at their 5′-termini were cloned into the pEU-E01-MCS plasmid (CellFree Sciences, Ehime, Japan), which is designed to produce mRNAs under the *SP6* promoter. This created the pEU-E01-3xFLAG-PNPLA1, pEU-E01-3xFLAG-PNPLA1(A34T), pEU-E01-3xFLAG-PNPLA1(A59V) and pEU-E01-3xFLAG-PNPLA1(E131X) plasmids.

**Lipid labelling assay.** HEK 293T cells were transfected with plasmids according to test group. Twenty-three hours and thirty minutes after transfection, medium was changed to DMEM (without FBS). After 30 min incubation, cells were labelled with 0.2 μCi [3-$^3$H]sphingosine (20 Ci mmol$^{-1}$; PerkinElmer Life Sciences, Waltham, MA, USA) for 6 h at 37 °C or 0.05 μCi [$^{14}$C]linoleic acid (55 mCi mmol$^{-1}$; American Radiolabeled Chemical, St Louis, MO, USA) for 4 h at 37 °C. In the case that cold linoleic acid (Sigma) was added to the medium, the addition was done at the same time as labelling. After washing with 1 ml of PBS, cells were suspended in 100 μl of PBS. Lipids were extracted by mixing 375 μl of chloroform/methanol/HCl (100:200:1, vol/vol/vol), 125 μl of chloroform and 125 μl of 1% KCl successively. Phases were then separated by centrifugation (20,000g, room temperature, 3 min).

The resulting organic (lower) phase was recovered, dried and dissolved in chloroform/methanol (2:1, vol/vol). Lipids were separated by normal-phase TLC (Silica Gel 60 TLC plates; Merck Millipore, Darmstadt, Germany). The following three TLC solvent systems were used sequentially: (i) chloroform/methanol/water (40:10:1, vol/vol/vol), developed to 2 cm from the bottom, dried and developed to 5 cm from the bottom; (ii) chloroform/methanol/acetic acid (47:2:0.5, vol/vol/vol), developed to the top; and (iii) hexane/diethylether/acetic acid (65:35:1, vol/vol/vol), developed to the top twice. Labelled lipids on TLCs were detected by spraying a fluorographic reagent (4 mg ml$^{-1}$ 2,5-diphenyl-oxazole in 2-methylnaphthalene/toluene (9:1, vol/vol)) and exposing to X-ray film at $-80\,^{\circ}\mathrm{C}$.

**Lipid analysis by LC-MS/MS.** Acylceramides containing sphingosine (EOS; esterified omega-hydroxyacyl-sphingosine) and TGs containing linoleic acid were quantified by LC-MS/MS. After washing with PBS, cells were suspended in 100 μl of PBS. Lipids were extracted by successive addition and mixing of 375 μl of chloroform/methanol/12 M formic acid (100:200:1, vol/vol/vol), 125 μl of chloroform and 125 μl of water. Phases were separated by centrifugation (9,000$g$, room temperature, 1 min). The protein aggregates lying between the water phase and the organic phase were recovered, and protein amounts were quantified using the Pierce BCA Protein Assay Kit (Thermo Fisher Scientific, Waltham, MA, USA). Lipids were recovered from the organic phase, dried and dissolved in 50 μl of chloroform/methanol (1:1, vol/vol). They were then resolved by ultra performance LC (UPLC) on a reverse-phase column (ACQUITY UPLC CSH C18 column; particle size 1.7 μm; inner diameter 2.1 mm; length 100 mm; Waters, Milford, MA, USA) at 55 °C, and detected by electrospray ionization (ESI) tandem triple quadrupole MS (Xevo TQ-S; Waters). The flow rate was 0.4 ml min$^{-1}$ in the binary gradient system using a mobile phase A (acetonitrile/water (3:2, vol/vol) containing 10 mM ammonium formate) and a mobile phase B (2-propanol/acetonitrile (9:1, vol/vol) containing 10 mM ammonium formate). The gradient steps were as follows: 0 min, 40% B; 0–18 min, linear gradient to 100% B; 18–23 min, 100% B; 23–23.1 min, linear gradient to 40% B; 23.1–25 min, 40% B. The injection volume was 5 μl. The ESI parameters were as follows: ion mode, positive; capillary voltage, 3.0 kV; sampling cone, 30 V; source offset, 50 V; desolvation temperature, 500 °C; desolvation gas flow, 1,000 l h$^{-1}$; cone gas flow, 150 l h$^{-1}$; nebulizer gas, 7.0 bar. Each acylceramide species containing sphingosine as the long-chain base component was detected by multiple reaction monitoring (MRM) mode by selecting the $m/z$ ($[M-H_2O+H]^+$ and $[M+H]^+$) of specific acylceramide species at Q1 and the $m/z$ 264.2 at Q3 and using the appropriate collision energy as described in Supplementary Table 2. For quantification, a standard curve was generated by serial dilutions of the C30:0 acylceramide EOS ($N$-(30-linoleoyloxy-triacontanoyl)-sphingosine; Matreya, Pleasant Gap, PA, USA). Each TG species containing linoleic acid was detected by MRM mode by selecting the $m/z$ $[M+NH_4]^+$ of specific TG species at Q1 and the $[M+H-297.2]^+$ (neutral loss of $[C18:2+NH_3]$) corresponding to diacylglycerol product ions at Q3 as described in Supplementary Table 3. Collision energy was set at 20 V. TGs were quantified using a standard curve plotted from serial dilutions of the TG 1-stearin-2-olein-3-linolein (Larodan Fine Chemicals AB, Malmo, Sweden). Data analysis and quantification were performed using MassLynx software (Waters).

**Gene knockdown.** The lentiviral vector pNS64 was constructed by modifying the restriction sites of pGFP-C-shLenti (OriGene Technologies, Rockville, MD, USA). The pNS72 plasmid encoding sh*PNPLA1* was constructed as follows. Oligo DNAs containing the *PNPLA1* shRNA target sequence (shPNPLA1-F and -R; Supplementary Table 1) were annealed and cloned into pAK1072, the vector for shRNA production under the U6 promoter, generating the pNS56 plasmid. The U6-sh*PNPLA1* region in the pNS56 was digested and two copies of the digested fragment were tandemly inserted into the pNS56 plasmid, generating pNS68 plasmid. The resulting total of three copies of U6-sh*PNPLA1* in the pNS56 plasmid were digested and cloned into the pNS64 vector, generating the pNS72 plasmid.

Twenty-four hours after seeding in a six-well plate ($2.0 \times 10^6$ cells per well), Lenti-X 293T cells were transfected with 1 μg of control shRNA vector (OriGene Technologies) or the pNS72 plasmid, together with 0.75 μg of the lentiviral packaging plasmid psPAX2 (Addgene, Cambridge, MA, USA) and 0.5 μg of the VSV-G envelope-expressing plasmid pMD2.G (Addgene). Medium was changed to fresh DMEM containing 10% FBS at 24 and 48 h after transfection, and the media collected at 48 and 72 h after transfection were pooled and centrifuged (1,500$g$, 4 °C, 10 min). The supernatant was centrifuged (50,000$g$, 4 °C, 2 h), and the resultant pellets were suspended in 300 μl of CnT prime Epidermal Keratinocyte Medium and used as the viral solution.

Human primary keratinocytes were seeded at $2.0 \times 10^4$ cells per well in a 12-well plate and cultured in CnT prime Epidermal Keratinocyte Medium for 24 h at 37 °C. The culture medium was replaced with 0.5 ml of the same medium but containing 25 μl of virus solution and 8 μg ml$^{-1}$ polybrene (Nacalai tesque, Kyoto, Japan). Six hours after incubation, the medium was changed to the one without the virus solution and polybrene, and incubated for 3 days. Differentiation was induced by incubating the infected cells with CnT-Prime 3D Barrier medium (CELLnTEC) for 7 days.

**Immunoblotting.** After separation by SDS-PAGE, proteins were electrotransferred to an Immobilon PVDF membrane (Millipore, Billerica, MA, USA). The membrane was incubated with blocking solution (5% skim milk in TBS-T (20 mM Tris-HCl (pH 7.5), 137 mM NaCl, and 0.05% Tween 20)) for 1 h at room temperature and then with anti-FLAG (M2; 1.85 μg ml$^{-1}$; Sigma; F3165) antibody in blocking solution for 1 h at room temperature. After washing with TBS-T three times, the membrane was incubated with an HRP-conjugated anti-mouse IgG F(ab′)$_2$ fragment (1:7,500 dilution; GE Healthcare Life Sciences, Little Chalfont, UK; NA9310) for 1 h at room temperature, followed by washing with TBS-T three times. Labelling was detected using Pierce ECL Western Blotting Substrate (Thermo Fisher Scientific).

**RT-PCR.** Total RNAs were isolated from cells using NucleoSpin RNA (Takara Bio). RT-PCR was performed using a PrimeScript One Step RT-PCR Kit (Takara Bio) and specific primer sets (for *PNPLA1*, PNPLA1-F2 and -R2; for *GAPDH*, GAPDH-F and -R; Supplementary Table 1), according to the manufacturer's protocols.

**In vitro acylceramide synthesis assay.** PNPLA1 was translated *in vitro* by a wheat germ cell-free system in the presence of liposomes using the ProteoLiposome Expression kit (CellFree Sciences). Liposomes were prepared as follows. Lipid solutions in chloroform (2 mg phosphatidylcholine (1-palmitoyl-2-oleoyl-*sn*-glycero-3-phosphocholine; Avanti Polar Lipids) with 20 μg TG (trilinolein; Sigma), 20 μg linoleoyl-CoA (Avanti Polar Lipids), and 20 μg C30:0 ω-hydroxyceramide ($N$-omega-hydroxytriacontanoyl-D-*erythro*-sphingosine; Matreya) as necessary) were dried in glass tubes. The dried lipid films were suspended in 80 μl of 1xSUB-AMIX SGC, incubated for 1 h at room temperature, and sonicated for 5 min.

Wild type and mutant *PNPLA1* mRNAs were transcribed from the pEU-E01-3xFLAG-PNPLA1, pEU-E01-3xFLAG-PNPLA1(A34T), pEU-E01-3xFLAG-PNPLA1(A59V) and pEU-E01-3xFLAG-PNPLA1(E131X) plasmids using SP6 RNA polymerase, according to the manufacturer's instructions. As a control, the pEU-E01-T1R1 plasmid in the kit was also used. The prepared mRNAs were then subjected to *in vitro* translation. Each translation mixture (5 μl of liposome, 6.25 μl of WEPRO7240, 0.1 μl of 20 mg ml$^{-1}$ creatine kinase, 3.75 μl of transcription mixture, 6.15 μl of 1xSUB-AMIX SGC, and 3.75 μl of water) was mixed in a 1.5 ml tube, transferred to the bottom of a well of 96-well plate containing 175 μl of 1xSUB-AMIX SGC to form bilayers, and incubated for 20 h at 15 °C. The resulting proteoliposomes were collected by centrifugation (20,000$g$, 4 °C, 10 min), washed with 1 ml of PBS three times, and suspended in Assay buffer (50 mM HEPES/NaOH (pH7.4), 150 mM NaCl, 10% glycerol). An acylceramide synthesis assay was then performed by incubating the proteoliposomes for 1 h at 37 °C. Lipid extraction and quantification of the C30:0 acylceramide EOS by LC-MS/MS were conducted as described above.

**Statistical analysis.** Values in the figure panels represent the means ± s.d.s of three or six independent experiments. Statistically significant differences were calculated using non-paired two-tailed Student's $t$-tests ($*P < 0.05$, $**P < 0.01$).

**Data availability.** All relevant data are available from the authors on request.

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

## Acknowledgements

This work was supported by funding from Advanced Research and Development Programs for Medical Innovation (AMED-CREST) (to A.K.) from Japan Agency for Medical Research and Development (AMED), by funding from Creation of Innovation Centers for Advanced Interdisciplinary Research Areas Program (to A.K.) from the Ministry of Education, Culture, Sports, Science and Technology of Japan, by a Grant-in-Aid for Scientific Research (A) 26251010 (to A.K.) from the Japan Society for the Promotion of Science (JSPS), and by a Grant-in-Aid for Young Scientists (A) 15H05589 (to Y.O.) from JSPS.

## Author contributions

Y.O. designed and performed the experiments and analysed the data. N.K. and S.N. performed the experiments. A.K. planned the project, designed the experiments and wrote the manuscript.

## Additional information

**Competing financial interests:** The authors declare no competing financial interests.

