## [Peer Review File · Nature Communications]

Reviewers' comments:

Reviewer #1 (expert in lipid metabolism)

Remarks to the Author:

In the manuscript by Ohno et al, the authors set out to identify the enzyme responsible for the ester bond formation between C18:2 FA and omega-hydroxyceramide to form omega-acylceramide. Using an overexpression based cellular labeling system, PNPLA1 was identified to be involved in omega-acylceramide generation. After carrying out multiple experiments for confirmation such as LC/MS measurements of acylceramides and use of autosomal recessive congenital ichthyosis related mutants of PNPLA1, the authors conclude that PNPLA1 works as a transacylase using TG as a donor of the substrate linoleic acid (C18:2 FA) to generate omega-acylceramide. However, the data presented are lacking at several aspects to support the authors' conclusions. In addition, the study lacks the molecular mechanism of how defective PNPLA1-induced loss of omega-acylceramides leads to ichthyosis. Below are some point by point comments:

Major points:

Is TG generation required for omega-acylceramide formation? If PNPLA1 is a transacylase, one would still expect to see a decrease in TG levels in Figures 2B and 5C. The involvement of DGAT2 and TG in the pathway is not obvious at all. How about the involvement of DGAT1: Is DGAT1 involved in omega-acylceramide generation? Overexpression effect can have non-specific activities therefore knock down or inhibition of DGAT1 and 2 effects on omega-acylceramide generating ability of PNPLA1 would be informative. Specifically, after the activation of PNPLA1 and ABHD5 co-expression (Figure 5B), leading to 16.6 fold increase in omega-acylceramide generation one might expect some decrease in TG levels if PNPLA1 was acting as a transacylase using the TG pool as a source of C18:2 FA. The data do not convincingly support the role of PNPLA1 as a transacylase.

The data suggest that PNPLA1 is involved in omega-acylceramide generation, in order to conclude that PNPLA1 is involved in the pathology, one needs to examine the effects of the PNPLA1 in a mouse model of ichthyosis

The mechanism of how ABHD5 activates PNPLA1 is not identified in the study at all. What is the expression profile of ABHD5 in skin? Does it overlap with PNPLA1? Do they physically associate? How does ABHD5 enhance PNPLA1 activity (changes in Km, Vmax)? Is the activity of PNPLA1 dependent on endogenous ABHD5? What determines the function of ABHD5 on regulating ATGL versus PNPLA1 activities?

Does microsomal PNPLA1 have in vitro activity towards generation of omega-acylceramide from omega-hydroxyceramide and TG (as C18:2 FA source) in the presence or absence of ABHD5? Similarly, do the mutant forms of PNPLA1 have reduced in vitro activities in a similar assay?

Other points:

Acylceramide levels in the presence of PNPLA1 shRNA should be measured?

Line 213- The data described refer to Figure 5b, not 5c. Please remove "c"

Reviewer #2 (expert in skin barrier)

Remarks to the Author:

The manuscript by Ohno et al., titled "Formation of the skin barrier lipid omega-O-acylceramide by the

ichthyosis gene PNPLA1" has been reviewed.

The authors expand upon their previously-developed cell system (that expresses genes for enzymes required for production of omega-hydroxyceramides) to investigate the potential role of a number of ichthyosis-causative genes in the formation of key epidermal barrier lipids, omega-acylceramides (omega-acylCer). Using this novel cell system, they present data to demonstrate requirement for PNPLA1 in omega-acylCer formation, as well as possible role of ABHD5 as a regulating step the formation of this critical epidermal barrier lipid. This is an important finding.

The study is original and of general interest due to the direct connection to important skin disorders (ichthyoses). Study is well-performed, including appropriate methodology and statistics, and the manuscript is well-written. The cell system used to study/produce the very hydrophobic acylCer species overcomes major hurdles involved in the study of the synthesis and metabolism of these critical epidermal lipids, and as such, is a significant advance for the field. The major biochemical findings are very timely. In addition, the demonstration of reduced acylCer formation by ichthyosis-causative mutants is also significant, as is the novel role for ABHD5 being a regulator of acylCer synthesis in the cell system. Conclusions appear valid, with appropriate use of statistics. References are appropriate.

Overall Suggestion:

Decrease emphasis on triglyceride formation/DGAT2 and ABHD5, and reduce data on these topics accordingly (details below).

Comments/Concerns:

Although the demonstration of increased acylCer formation in the presence of over-expressed DGAT2 is interesting, it is not a key finding of this study. As such, the demonstration of increased triglyceride formation by DGAT2 (but not PNPLA1) appears to be a distraction. Therefore, authors should reduce discussion on this point, and consider moving Fig. 2 to supplemental information/data, and focus manuscript on key PNPLA1 data/findings.

Although the identification of a novel role for ABHD5 as regulator of acylCer formation is of significant interest, the selective increase(s) in acylCer with specific N-acyl chain lengths (and mono-unsaturation), creates more questions than are answered by the authors, and as such, distract from the main point. Given that the mechanism by which ABHD5 enhances PHPLA1 activity/acylCer formation remains unresolved, authors should consider including only those data that are critical to current PNPLA1-centric manuscript. Also, the negative data in figure 5c and 5d are not particularly insightful, and should be removed or relegated to supplementary data.

Additional bands (migrating faster than the GlcCer bands) in ABHD5- and LIPN-transfected cell systems (Fig. 1b) should be addressed.

Thank you very much for the reviews of our manuscript (MS# NCOMMS-16-08436) and the useful comments. We have performed additional experiments and changed the text and figures accordingly. The following are our itemized responses to the reviewers.

REVIEWER 1

Comment 1: “*Is TG generation required for omega-acylceramide formation? If PNPLA1 is a transacylase, one would still expect to see a decrease in TG levels in Figures 2B and 5C. The involvement of DGAT2 and TG in the pathway is not obvious at all. How about the involvement of DGAT1: Is DGAT1 involved in omega-acylceramide generation? Overexpression effect can have non-specific activities therefore knock down or inhibition of DGAT1 and 2 effects on omega-acylceramide generating ability of PNPLA1 would be informative. Specifically, after the activation of PNPLA1 and ABHD5 co-expression (Figure 5B), leading to 16.6 fold increase in omega-acylceramide generation one might expect some decrease in TG levels if PNPLA1 was acting as a transacylase using the TG pool as a source of C18:2 FA. The data do not convincingly support the role of PNPLA1 as a transacylase.*”

Response

In the revised manuscript, we have added an *in vitro* result, which shows that PNPLA1 catalyzes acylceramide production using TG as a substrate (new Fig. 5). This result clearly demonstrated that PNPLA1 is a *bona fide* transacylase for acylceramide generation using TG as a substrate. Thus, this new finding proves the authenticity of our results obtained from the cell-based assay, which showed that an increase in TG by DGAT2 overproduction stimulated acylceramide production. Consistent with our results, it has been reported that *Dgat2* knockout mice exhibit a skin barrier defect phenotype (Stone SJ *et al. J. Biol. Chem.*, 279, 11767-11776; 2004). We speculate that the decreased TG causes impairment of acylceramide synthesis, leading to the skin barrier defect. However, the role of DGAT2 in skin barrier formation has not been verified experimentally, so we left this notion as only a possibility in the Discussion, with care not to overstate it. In contrast to *Dgat2* knockout mice exhibiting a skin barrier defect phenotype, the skin of *Dgat1* knockout mice is normal. Therefore, the involvement of DGAT1 in acylceramide production is unlikely.

The reviewers expected a decrease in TG levels by overproduction of PNPLA1 in our previous Figs. 2B and 5C, if PNPLA1 was a transacylase using TG as a substrate. However, this seeming discrepancy was due to the cell system we used, where only PNPLA1 was overproduced. Acylceramide synthesis also requires the presence of ω -hydroxyceramide, and production of this was achieved by co-expression of the fatty acid elongase ELOVL4,

ceramide synthase CERS3, and ω -hydroxylase CYP4F22 with PNPLA1. Therefore, overproduction of PNPLA1 alone could not cause the transacylation. Accordingly, TG levels were not decreased. In those experiments we examined the possibility that PNPLA1 acted as a TG hydrolase. This was why we simply overproduced PNPLA1 alone.

Comment 2: “*The data suggest that PNPLA1 is involved in omega-acylceramide generation, in order to conclude that PNPLA1 is involved in the pathology, one needs to examine the effects of the PNPLA1 in a mouse model of ichthyosis.*”

Response

Such data are presented by Dr. Murakami and colleagues (Hirabayashi *et al.*), and their paper is submitted side by side with ours. Their data indeed showed that deficiency of *Pnpla1* leads to an ichthyosis phenotype. The results obtained from these different approaches (cell-based and biochemical assays by us and *in vivo* analyses by Dr. Murakami's group) support each other, and contribute to an in-depth understanding of the roles of PNPLA1 in acylceramide production. We hope that publication of the two manuscripts side-by-side in *Nature Communications* will be a highly valuable showcase to propagate this new understanding of the molecular mechanisms behind skin barrier formation.

Comment 3: “*The mechanism of how ABHD5 activates PNPLA1 is not identified in the study at all. What is the expression profile of ABHD5 in skin? Does it overlap with PNPLA1? Do they physically associate? How does ABHD5 enhance PNPLA1 activity (changes in Km, Vmax)? Is the activity of PNPLA1 dependent on endogenous ABHD5? What determines the function of ABHD5 on regulating ATGL versus PNPLA1 activities?*”

Response

We agree with your comments that the regulatory mechanism of ABHD5 toward PNPLA1 activity is unclear at present. In accordance with the second reviewer's comment that we should focus on PNPLA1 in this manuscript, we have removed the description of ABHD5. Instead, we have added new results for PNPLA1, including the *in vitro* acylceramide synthesis activity of wild type (new Fig. 5b) and mutant PNPLA1 (new Fig. 6d), and LC-MS analysis of keratinocytes with *PNPLA1* gene knockdown (new Fig. 2c).

Comment 4: “*Does microsomal PNPLA1 have in vitro activity towards generation of omega-acylceramide from omega-hydroxyceramide and TG (as C18:2 FA source) in the*

presence or absence of ABHD5? Similarly, do the mutant forms of PNPLA1 have reduced in vitro activities in a similar assay?"

Response

We performed an *in vitro assay* using PNPLA1 translated by a wheat germ cell-free translation system in the presence of liposomes. Acylceramide was indeed generated by PNPLA1 in the presence of TG but not of linoleoyl-CoA (new Fig. 5b). We also examined the activity of the mutant forms of PNPLA1 and revealed that they had reduced or no activities (new Fig. 6d).

Comment 5: *"Acylceramide levels in the presence of PNPLA1 shRNA should be measured?"*

Response

We measured acylceramide levels in the knockdown experiments and found decreases as expected (new Fig. 2c).

Comment 6: *"Line 213- The data described refer to Figure 5b, not 5c. Please remove "c"."*

Response

Thank you for pointing out our mistake.

REVIEWER 2

Comment 1: *"Although the demonstration of increased acylCer formation in the presence of over-expressed DGAT2 is interesting, it is not a key finding of this study. As such, the demonstration of increased triglyceride formation by DGAT2 (but not PNPLA1) appears to be a distraction. Therefore, authors should reduce discussion on this point, and consider moving Fig. 2 to supplemental information/data, and focus manuscript on key PNPLA1 data/findings."*

Response

We agree that the involvement of TG/DGAT2 was unclear in the original manuscript and appreciate your suggestion. However, in the course of the revision we obtained *in vitro* data that showed PNPLA1 synthesized acylceramide using TG as a substrate. From this result, we believe that the involvement of TG in acylceramide production is now more apparent. Therefore, we have left the description of TG in the main text of the revised manuscript.

However, regarding DGAT2, we did not perform knockdown/knockout analyses. Therefore, we kept our notion about the function of DGAT2 in skin barrier formation (*i.e.*, that DGAT2 supplies TG, the substrate of acylceramide synthesis) as only a possibility in the Discussion, with care not to overstate this idea.

Comment 2: *“Although the identification of a novel role for ABHD5 as regulator of acylCer formation is of significant interest, the selective increase(s) in acylCer with specific N-acyl chain lengths (and mono-unsaturation), creates more questions than are answered by the authors, and as such, distract from the main point. Given that the mechanism by which ABHD5 enhances PHPLA1 activity/acylCer formation remains unresolved, authors should consider including only those data that are critical to current PNPLA1-centric manuscript. Also, the negative data in figure 5c and 5d are not particularly insightful, and should be removed or relegated to supplementary data.”*

Response

Thank you for your constructive suggestion. We have removed the data on ABHD5 and have focused on PNPLA1 in the revised manuscript.

Comment 3: *“Additional bands (migrating faster than the GlcCer bands) in ABHD5- and LIPN-transfected cell systems (Fig. 1b) should be addressed.”*

Response

As you noticed, unknown bands were observed in that experiment, when ABHD5 or LIPN were expressed. However, the appearance of these bands was not reproduced in other experiments. Therefore, we did not analyze them further. We have now provided this explanation in the figure legend of the revised manuscript.

We are grateful to the reviewers for recognizing the significance of our study and for providing insightful and constructive comments.

Thank you very much for the reviews of our manuscript (MS# NCOMMS-16-08436) and the useful comments. We have performed additional experiments and changed the text and figures accordingly. The following are our itemized responses to the reviewers.

REVIEWER 1

Comment 1: “*Is TG generation required for omega-acylceramide formation? If PNPLA1 is a transacylase, one would still expect to see a decrease in TG levels in Figures 2B and 5C. The involvement of DGAT2 and TG in the pathway is not obvious at all. How about the involvement of DGAT1: Is DGAT1 involved in omega-acylceramide generation? Overexpression effect can have non-specific activities therefore knock down or inhibition of DGAT1 and 2 effects on omega-acylceramide generating ability of PNPLA1 would be informative. Specifically, after the activation of PNPLA1 and ABHD5 co-expression (Figure 5B), leading to 16.6 fold increase in omega-acylceramide generation one might expect some decrease in TG levels if PNPLA1 was acting as a transacylase using the TG pool as a source of C18:2 FA. The data do not convincingly support the role of PNPLA1 as a transacylase.*”

Response

In the revised manuscript, we have added an *in vitro* result, which shows that PNPLA1 catalyzes acylceramide production using TG as a substrate (new Fig. 5). This result clearly demonstrated that PNPLA1 is a *bona fide* transacylase for acylceramide generation using TG as a substrate. Thus, this new finding proves the authenticity of our results obtained from the cell-based assay, which showed that an increase in TG by DGAT2 overproduction stimulated acylceramide production. Consistent with our results, it has been reported that *Dgat2* knockout mice exhibit a skin barrier defect phenotype (Stone SJ *et al. J. Biol. Chem.*, 279, 11767-11776; 2004). We speculate that the decreased TG causes impairment of acylceramide synthesis, leading to the skin barrier defect. However, the role of DGAT2 in skin barrier formation has not been verified experimentally, so we left this notion as only a possibility in the Discussion, with care not to overstate it. In contrast to *Dgat2* knockout mice exhibiting a skin barrier defect phenotype, the skin of *Dgat1* knockout mice is normal. Therefore, the involvement of DGAT1 in acylceramide production is unlikely.

The reviewers expected a decrease in TG levels by overproduction of PNPLA1 in our previous Figs. 2B and 5C, if PNPLA1 was a transacylase using TG as a substrate. However, this seeming discrepancy was due to the cell system we used, where only PNPLA1 was overproduced. Acylceramide synthesis also requires the presence of ω -hydroxyceramide, and production of this was achieved by co-expression of the fatty acid elongase ELOVL4,

ceramide synthase CERS3, and ω -hydroxylase CYP4F22 with PNPLA1. Therefore, overproduction of PNPLA1 alone could not cause the transacylation. Accordingly, TG levels were not decreased. In those experiments we examined the possibility that PNPLA1 acted as a TG hydrolase. This was why we simply overproduced PNPLA1 alone.

Comment 2: “*The data suggest that PNPLA1 is involved in omega-acylceramide generation, in order to conclude that PNPLA1 is involved in the pathology, one needs to examine the effects of the PNPLA1 in a mouse model of ichthyosis.*”

Response

Such data are presented by Dr. Murakami and colleagues (Hirabayashi *et al.*), and their paper is submitted side by side with ours. Their data indeed showed that deficiency of *Pnpla1* leads to an ichthyosis phenotype. The results obtained from these different approaches (cell-based and biochemical assays by us and *in vivo* analyses by Dr. Murakami's group) support each other, and contribute to an in-depth understanding of the roles of PNPLA1 in acylceramide production. We hope that publication of the two manuscripts side-by-side in *Nature Communications* will be a highly valuable showcase to propagate this new understanding of the molecular mechanisms behind skin barrier formation.

Comment 3: “*The mechanism of how ABHD5 activates PNPLA1 is not identified in the study at all. What is the expression profile of ABHD5 in skin? Does it overlap with PNPLA1? Do they physically associate? How does ABHD5 enhance PNPLA1 activity (changes in Km, Vmax)? Is the activity of PNPLA1 dependent on endogenous ABHD5? What determines the function of ABHD5 on regulating ATGL versus PNPLA1 activities?*”

Response

We agree with your comments that the regulatory mechanism of ABHD5 toward PNPLA1 activity is unclear at present. In accordance with the second reviewer's comment that we should focus on PNPLA1 in this manuscript, we have removed the description of ABHD5. Instead, we have added new results for PNPLA1, including the *in vitro* acylceramide synthesis activity of wild type (new Fig. 5b) and mutant PNPLA1 (new Fig. 6d), and LC-MS analysis of keratinocytes with *PNPLA1* gene knockdown (new Fig. 2c).

Comment 4: “*Does microsomal PNPLA1 have in vitro activity towards generation of omega-acylceramide from omega-hydroxyceramide and TG (as C18:2 FA source) in the*

presence or absence of ABHD5? Similarly, do the mutant forms of PNPLA1 have reduced in vitro activities in a similar assay?"

Response

We performed an *in vitro assay* using PNPLA1 translated by a wheat germ cell-free translation system in the presence of liposomes. Acylceramide was indeed generated by PNPLA1 in the presence of TG but not of linoleoyl-CoA (new Fig. 5b). We also examined the activity of the mutant forms of PNPLA1 and revealed that they had reduced or no activities (new Fig. 6d).

Comment 5: *"Acylceramide levels in the presence of PNPLA1 shRNA should be measured?"*

Response

We measured acylceramide levels in the knockdown experiments and found decreases as expected (new Fig. 2c).

Comment 6: *"Line 213- The data described refer to Figure 5b, not 5c. Please remove "c"."*

Response

Thank you for pointing out our mistake.

REVIEWER 2

Comment 1: *"Although the demonstration of increased acylCer formation in the presence of over-expressed DGAT2 is interesting, it is not a key finding of this study. As such, the demonstration of increased triglyceride formation by DGAT2 (but not PNPLA1) appears to be a distraction. Therefore, authors should reduce discussion on this point, and consider moving Fig. 2 to supplemental information/data, and focus manuscript on key PNPLA1 data/findings."*

Response

We agree that the involvement of TG/DGAT2 was unclear in the original manuscript and appreciate your suggestion. However, in the course of the revision we obtained *in vitro* data that showed PNPLA1 synthesized acylceramide using TG as a substrate. From this result, we believe that the involvement of TG in acylceramide production is now more apparent. Therefore, we have left the description of TG in the main text of the revised manuscript.

However, regarding DGAT2, we did not perform knockdown/knockout analyses. Therefore, we kept our notion about the function of DGAT2 in skin barrier formation (*i.e.*, that DGAT2 supplies TG, the substrate of acylceramide synthesis) as only a possibility in the Discussion, with care not to overstate this idea.

Comment 2: *“Although the identification of a novel role for ABHD5 as regulator of acylCer formation is of significant interest, the selective increase(s) in acylCer with specific N-acyl chain lengths (and mono-unsaturation), creates more questions than are answered by the authors, and as such, distract from the main point. Given that the mechanism by which ABHD5 enhances PHPLA1 activity/acylCer formation remains unresolved, authors should consider including only those data that are critical to current PNPLA1-centric manuscript. Also, the negative data in figure 5c and 5d are not particularly insightful, and should be removed or relegated to supplementary data.”*

Response

Thank you for your constructive suggestion. We have removed the data on ABHD5 and have focused on PNPLA1 in the revised manuscript.

Comment 3: *“Additional bands (migrating faster than the GlcCer bands) in ABHD5- and LIPN-transfected cell systems (Fig. 1b) should be addressed.”*

Response

As you noticed, unknown bands were observed in that experiment, when ABHD5 or LIPN were expressed. However, the appearance of these bands was not reproduced in other experiments. Therefore, we did not analyze them further. We have now provided this explanation in the figure legend of the revised manuscript.

We are grateful to the reviewers for recognizing the significance of our study and for providing insightful and constructive comments.

REVIEWERS' COMMENTS:

Reviewer #1 (Remarks to the Author):

Most of the concerns were addressed, however a few comments/concerns remain as follows

The authors provide data (Fig 5) from in vitro activity assay for acylceramide production using proteoliposomes to show the transacylase activity of in vitro translated PNPLA1. While incubation of PNPLA1 with TG and omega-OH-Cer results in generation of acylceramide, there is significant acylceramide production in the presence of C18:2-CoA and omega-OH-Cer, suggesting that PNPLA1 has C18:2-CoA dependent acyltransferase activity in addition to the proposed transacylase property. This observation should be mentioned in results and be discussed. Moreover, the conversion of TG to DAG should be demonstrated in the in vitro activity assay in order to conclude that PNPLA1 is a bona fide transacylase.

According to the data presented in Fig 5b, there is significant acylceramide generation in the presence of C18:2-CoA and omega-OH-Cer (3rd bar from the last) suggesting that PNPLA1 also has CoA-dependent acyltransferase activity. This data should be mentioned and discussed (as mentioned for point 1 above).

Authors performed the required experiment of downregulating PNPLA1 and measuring acylceramide.

Reviewer #2 (Remarks to the Author):

The Authors have addressed each of this Reviewer's concerns, and the manuscript has been significantly improved. Therefore, this Reviewer has no additional comments for these Authors.

We thank you very much for the reviews of our manuscript (MS# NCOMMS-16-08436B) and the useful comments. We have added a figure accordingly and below is our response to the comment from reviewer #1.

REVIEWER 1

Comment 1: *“The authors provide data (Fig 5) from in vitro activity assay for acylceramide production using proteoliposomes to show the transacylase activity of in vitro translated PNPLA1. While incubation of PNPLA1 with TG and omega-OH-Cer results in generation of acylceramide, there is significant acylceramide production in the presence of C18:2-CoA and omega-OH-Cer, suggesting that PNPLA1 has C18:2-CoA dependent acyltransferase activity in addition to the proposed transacylase property. This observation should be mentioned in results and be discussed. Moreover, the conversion of TG to DAG should be demonstrated in the in vitro activity assay in order to conclude that PNPLA1 is a bone fide transacylase. According to the data presented in Fig 5b, there is significant acylceramide generation in the presence of C18:2-CoA and omega-OH-Cer (3rd bar from the last) suggesting that PNPLA1 also has CoA-dependent acyltransferase activity. This data should be mentioned and discussed (as mentioned for point 1 above).”*

Response

As already described in the manuscript, low levels of acylceramides were produced by PNPLA1 in the presence of ω -hydroxyceramide alone (without exogenous TG). This is probably due to the supply of TG from the wheat germ lysates used in the cell-free translation system, as we confirmed by LC-MS analysis. Inclusion of linoleoyl-CoA in the proteoliposomes containing PNPLA1 and ω -hydroxyceramide did not cause further increase in acylceramide levels. To clarify this, we added "n.s." (not significant) in the figure. Therefore, we cannot conclude that PNPLA1 exhibits C18:2-CoA dependent acyltransferase activity in addition to transacylase activity. DAG was indeed generated by PNPLA1 only in the presence of ω -hydroxyceramide and TG. We added this result to the newly added Supplementary Figure 1.

We are grateful to the reviewers for recognizing the significance of our study and for providing insightful and constructive comments. We believe that our manuscript has been greatly improved by these revisions and hope that it is now acceptable for publication in *Nature Communications*.